# Chaos in the vicinity of a singularity in the Three-Body Problem: The equilateral triangle experiment in the zero angular momentum limit

Hugo D. Parischewsky[1*], Gustavo Ceballos[1], Alessandro A. Trani[2,3] and Nathan W. C. Leigh[1,4]

**1** Departamento de Astronomía, Facultad de Ciencias Físicas y Matemáticas, Universidad de Concepción, Chile, Avenida Esteban Iturra s/n Casilla 160-C
**2** Department of Earth Science and Astronomy, College of Arts and Sciences, The University of Tokyo, 3-8-1 Komaba, Meguro-ku, Tokyo 153-8902, Japan
**3** Okinawa Institute of Science and Technology, 1919-1 Tancha, Onna-son, Okinawa 904-0495, Japan
**4** Department of Astrophysics, American Museum of Natural History, New York, NY 10024, USA
* hugo.parischewsky.z@gmail.com

October 26, 2021

## Abstract

We present numerical simulations of the gravitational three-body problem, in which three particles lie at rest close to the vertex of an equilateral triangle. In the unperturbed problem, the three particles fall towards the center of mass of the system to form a three-body collision, or singularity, where the particles overlap in space and time. By perturbing the initial positions of the particles, we are able to study chaos in the vicinity of the singularity. Here we cover both the singular region close to the unperturbed configuration and the binary-single scattering regime when one side of the triangle is very short compared to the other two. We make phase space plots to study the regular and ergodic subsets of our simulations and compare them with the outcomes expected from the statistical escape theory of the three-body problem. We further provide fits to the ergodic subset to characterize the properties of the left-over binaries. We identify the discrepancy between the statistical theory and the simulations in the regular subset of interactions, which only exhibits weak chaos. As we decrease the scale of the perturbations in the initial positions, the phase space becomes entirely dominated by regular interactions, according to our metric for chaos. Finally, we show the effect of general relativity corrections by simulating the same scenario with the inclusion of post-Newtonian corrections to the equations of motion.

# 1   Introduction

The gravitational two-body problem was solved analytically by Newton in his *PhilosophiæNaturalis Principia Mathematica*. His work made it possible to predict exactly the positions and velocities of two self-gravitating point masses at any point in the future, given their initial masses, positions and velocities.

After Newton, generations of mathematicians and physicists tried to obtain an equivalently

elegant solution for the three-body problem, without success. It was eventually Poincaré [1] who showed that such a solution would be impossible since the general three-body problem is an example of chaos in nature.

Many centuries passed with little further progress, but the introduction of computer simulations in the late 1900's by Aarseth and collaborators [2–5] re-vitalised popularity in the three-body problem, allowing researchers to confront their analytic models directly with computer simulations. This allowed for the development of analytic approximations for regions of phase space that are regular (i.e., the interactions are prompt and never enter a long-lived resonant state; see [6] and the text below for a more detailed definition). However, only recently the probabilistic theories of the three-body problem have begun to be explored [7–13].

A prerequisite to having a chaotic three-body interaction is that the system enters a long-lived resonant state, where all three particles undergo numerous close approaches before the interaction ends. If the interaction ends promptly, with the incoming single star interacting with the initial binary components only once before the interaction ends, the interaction is said to be regular and analytic solutions can typically be found e.g. [14–16].

There are two main states for the time evolution of a chaotic three-body system in a resonant state. The first state consists of a hierarchy, which is composed of a temporary close binary system and a temporary single star going on a prolonged excursion with a total energy approaching zero but remaining negative. The time evolution is such that the system continually breaks apart into such a hierarchy, with the temporary single sometimes going on long-lived and sometimes short-lived excursions. The second state is such that all three particles are in approximate energy equipartition, leading to a fast and chaotic exchange of energy and angular momentum e.g. [17, 18]. These "scramble states" are needed for the system to become chaotic, such that the particles lose all memory of their initial conditions, recalling just the total energy, total angular momentum and the particle masses. The time evolution continues in this way, inter-changing between these two states chaotically. Eventually, if all bodies are point particles, the interaction ends with one of the particles being ejected with a positive total energy and a finite velocity at spatial infinity [9].

Chaos can be defined as occurring when small perturbations to the initial conditions results in different macroscopic outcomes (e.g., which of the three particles is the one to be ejected). The extreme sensitivity to the initial conditions manifests itself through the exponential growth of small perturbations that can exhibit unpredictable and divergent behaviour, yielding completely different outcomes for nearly identical sets of initial conditions [19]. This implies numerical and physical consequences [20] and, as already mentioned, points to the need to develop a probabilistic theory for chaotic regions of phase space [10, 11, 21, 22].

This extreme sensitivity to the initial conditions can also result in diverging trajectories through phase space purely due to the accumulation of numerical errors, which act as small perturbations to the system at each time-step, effectively mimicking and amplifying the effects of chaos. Consequently, a high degree of accuracy and precision is needed over long-time-scale integrations, to ensure that the simulated solutions are correct, and to remove the concern of numerical errors mimicking the effects of chaos. The development of sophisticated gravity integrators has proved essential by reducing the accumulation of numerical errors, and guaranteeing the reliability of orbital integrations over long periods of times (i.e., many orbital periods) [20, 23–25]. Most dynamical systems display some aspect of chaos, including solar system small bodies e.g. [26–28], small stellar systems e.g. [6, 10, 11, 29–33], star clusters e.g. [34, 35], galaxies e.g. [36], and so on.

In this paper, our focus is to study chaos in the vicinity of a singularity in the three-body

problem. The singularity we consider is a three-body collision at the system centre of mass, which occurs when the three particles are released from rest, each initially at the vertex of a perfect equilateral triangle. This creates a singularity in the gravitational acceleration, since the particles overlap in both space and time upon reaching the system centre of mass for this idealized initial configuration. By perturbing the planar equilateral triangle and releasing the particles from rest, they will arrive at the system centre of mass at nearly the same time, but slightly offset. In this way, we can probe the interaction outcomes directly in the vicinity of the singularity, resolving it down to very small spatial scales. Our experiments are designed to be entirely planar, with each system composed of three point-particles with equal masses, each located initially at a randomly sampled position for the corresponding vertex for that particle. This experiment is in the zero angular momentum limit, since there is no angular momentum for any configuration initially at rest.

In Section 2, we introduce a number of key definitions used throughout this paper to describe chaotic three-body interactions. In Section 3, we present our methods and experimental set-up, including justification for our choice of gravity integrator TSUNAMI [37] and the issue of regularization, the initial conditions, the number of simulations performed, and so on. In Section 4 we present the theoretical expressions used to compare analytic theory with our numerical scattering experiments. In Section 5, we present the results of our experiments, with a focus on comparing the simulated distributions for the final binary semi-major axis (i.e., the inverse binding energy distribution), eccentricity, single star escaper velocity, and the total interaction lifetimes to the theoretical expectations. In Section 6, we summarise our results and discuss their implications for the three-body problem, chaos in the vicinity of a singularity and if such experiments can be used to characterise gravity in the quantum regime. Finally, we discuss the properties of binaries formed via three-body interactions of isolated single stars in dense isotropic stellar systems.

## 2 Quantifying Chaos

In this section we introduce several key definitions used throughout the paper to describe chaotic interactions.

We introduce the concept of ergodicity, its relation to Lévy flights and their combined implications to defining a chaotic interaction in the three body problem. As already discussed, separating three-body interactions into ergodic and regular subsets is critical for a proper comparison between the simulated data and theoretical predictions. The regular subset tends to correspond to prompt interactions, for which analytic methods have already been developed to quantify the outcome properties as a function of the initial conditions. The ergodic subset tends to correspond to longer-lived resonant interactions, and require a probabilistic theory to quantify the outcome properties as a function of the initial conditions.

### 2.1 Ergodicity

When comparing simulations of three-body interactions to theoretical predictions, it is crucial to first define a criterion for when the interaction formally becomes a chaotic one. This is typically done by quantifying the amount of time all three particles have comparable energies and are in an approximate state of energy equipartition. If this criterion can be achieved multiple times, the system gradually loses memory of its initial conditions (i.e., the particles no longer remember from where they originated in the initial phase space), leaving the system only knowing the total

interaction energy, the total angular momentum and the particle masses. Stone and Leigh [11] found that if the system enters two or more such states, which the authors term "scrambles", then this removes most of the regular subset, leaving beyond only the ergodic subset of the interactions. More recently, other authors have begun to explore more sophisticated methods for defining when an interaction formally becomes ergodic e.g. [13, 38].

We also point out that longer integration times for the simulations tend to correlate with increased error accumulation. As already discussed, this can mimic the effects of chaos. Hence, one must be careful to manage the errors properly in order to identify the subset of interactions that should correspond to the true end state of the system (i.e., what nature would produce). Our methods for managing error accumulation in the simulations are described in detail in Section 3.

## 2.2 Lévy flights

Lévy flights correspond to very long excursions of one of the particles, where the particle has nearly zero but still negative total energy and remains barely bound to the system. These correlate with very long integration times, since the duration of these excursions can be very long e.g [39, 40].

The nature of Lévy flights remains an open research question. Lévy flights are a class of random walks with a heavy-tailed distribution. In our context, a Lévy flight is an extremely long excursion following a fast chaotic interaction (see [39–41]), whereas interactions without exceedingly long Lévy flights tend to follow an exponential drop-off in the cumulative interaction lifetimes, which can be described using a half-life formalism (see [13, 38–40] for a more in-depth discussion of the relevant physics).

The importance of Lévy flights for defining chaotic interactions is still not completely understood. For example, consider a three-body interaction that undergoes a very long excursion or Lévy flight with the energy of the single being very close to zero. The interaction ends promptly at the end of the Lévy flight, with the single interacting once with the temporary binary followed by a prompt ejection and the end of the interaction. If this experiment is repeated with minute perturbations to the initial conditions, one can imagine that these simulations, grouped very closely together in phase space, will all end with a prompt ejection of the returning single, only altering the properties of the ejected single and the left-over binary very slightly. This would, at least in principle, introduce a small regular subset into three-body interactions that were already defined as being chaotic, having achieved our chosen criterion for ergodicity. These subsets should occupy sufficiently small phase space volumes, however, that they should constitute $\ll 1\%$ of our chaotic subset.

## 2.3 Scramble cut

A scramble is defined as a temporary state in which all three particles have comparable energies and are not in a hierarchical configuration. We define $N_S$ as the number of scrambles, which is the number of times the system has entered into such a temporary state of approximate equipartition. If the system achieves approximate energy equipartition, it can lose all memory of the initial conditions and become ergodic, retaining only memory of the particle masses, the total interaction energy and the total angular momentum. Stone and Leigh [11] showed that two or more such scrambles are a sufficient criterion to remove most of the regular regions of phase space, leaving only the ergodic subset. In this paper, if a three-body interaction satisfies the criterion $N_S \geq 4$, where $N_S$ is the number of scrambles, it is defined to be a chaotic or ergodic interaction. After applying this cut, we find that 82.1% of our simulations correspond to the chaotic region and

17.9% to the regular region.

## 3   Method

In this section, we describe the numerical simulations used to perform all numerical experiments conducted in this paper. We first present TSUNAMI, the gravity integrator we use to compute the time evolution of our three body systems, before moving on to describing how we define our initial conditions and set up our experiments.

### 3.1   N-body Integrator

We use TSUNAMI [37], a few-body code ideally suited to follow the time evolution of strongly interacting systems. TSUNAMI manages the computations using sophisticated regularization techniques to achieve the required accuracy and precision, which are needed to properly model very close approaches between particles.

   TSUNAMI implements a leapfrog scheme derived from a time-transformed Hamiltonian, using a combination of the logH and TTL schemes [42,43]. Here we use only the logH formulation. Because the leapfrog scheme is accurate only to 2nd order, we increase the accuracy of the integration by employing a Bulirch-Stoer extrapolation scheme [44] with a tolerance of $10^{-13}$. TSUNAMI also includes the post-Newtonian terms of order 1, 2 and 2.5 [45]. The details and performance of TSUNAMI will be presented in a following paper (Trani et. al, in preparation).

### 3.2   Initial conditions

In the following sub-sections, we describe how the initial conditions for our suites of numerical simulations are defined.

#### 3.2.1   Unperturbed initial configuration

The initial conditions are generated by perturbing the positions of the particles from the unperturbed triangular configuration. The unperturbed triangular configuration is described here. We label each particle with the numbers 1,2,3, which also indicate the side of the triangle that each particle opposes. The initial, unperturbed positions for the particles 1 and 2 are $P_1 = (x_1, y_1, z_1) = (1.5, 1.5, 0)$ and $P_2 = (x_2, y_2, z_2) = (2.5, 1.5, 0)$. Note that the triangle side is 1 au. This represents a perfect equilateral triangle, where each side will have a length $s_\triangle = ((x_1 - x_2)^2 - (y_1 - y_2)^2)^{0.5}$ in Cartesian coordinates. The initial position of particle $P_3$ was then calculated as:

$$x_3 \ = \ x_1 + \frac{s_\triangle}{2} \tag{1}$$

$$y_3 \ = \ y_2 + \left( s_\triangle^2 - \left( \frac{s_\triangle}{2} \right)^2 \right)^{0.5} \tag{2}$$

we set $z = 0$, and focus on purely 2D coplanar configurations.

#### 3.2.2   Perturbed initial configurations

We perturb the initial positions of the particles in the following way. The position of every particle is chosen within a radius of size $r_{\text{gen}}$ from the vertex, as shown in Figure 1. First, we randomly

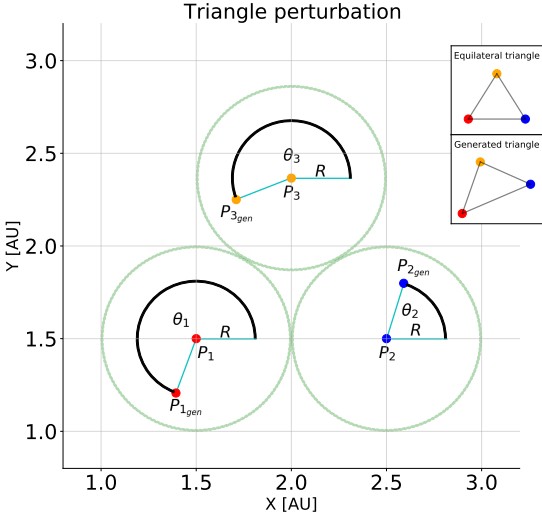

Figure 1: Scheme of the initial setup. The equilateral triangle vertices are labeled as $P_1$, $P_2$, and $P_3$. The generated particles corresponding to the perturbed triangle are labeled as $P_{1_\text{gen}}$, $P_{2_\text{gen}}$, and $P_{3_\text{gen}}$. The green circles correspond to the boundaries of each generated particle ($r_\text{gen} = 0.5\,\text{au}$). The insets in the upper right show the equilateral triangle and the perturbed version.

draw a perturbation radius $r$ within a circle of $r_\text{gen}$. Since $r$ represents a radial perturbation, we randomly sample it from a distribution uniform in $r_\text{gen}^2$. The position of each particle is perturbed by the same amount $r$, but with a radial direction that is independently drawn for each particle. Specifically, the direction of the perturbation is set by the angles $\theta_1$, $\theta_2$ and $\theta_3$ for particle 1, 2 and 3. The angles are sampled from a uniform distribution in the range $(0, 2\pi)$.

We generate three different sets of simulations with different perturbation magnitudes, in order to increase sampling in the vicinity of the singularity and to re-perform our calculations with post-Newtonian corrections turned on. To explore the entire phase space, we first sample each generated particle within a $r_\text{gen} = 0.5\,\text{au}$ circle positioned at the vertex of the opposing side of the triangle. To explore the vicinity of the singularity we increase our sampling resolution and use instead $r_\text{gen} = 10^{-7}\,\text{au}$.

We assume identical point particles and purely Newtonian forces for **Set A** and **Set B**. In **Set A** we assumes a vertex radius of $0.5\,\text{au}$, while for **Set B**, the sampling is performed using a vertex radius of $10^{-7}$ au.

We re-run the same simulations in **Set C** but with post-Newtonian corrections turned on. These are required very close to the singular regions in the three-body problem, in order to avoid non-physical solutions (e.g., escape velocities for the single star that exceed the velocity of light). This requires the introduction of a critical radius to define the conditions for a merger to occur, which we will come back to in more detail below. The initial particle velocities are always zero and the particle masses are all set to $1\,\text{M}_\odot$.

Table 1 shows the number of simulations performed for each set (Column 2), the particle radii (Column 3), and our choices for the sampling radii $r_\text{gen}$ (Column 4).

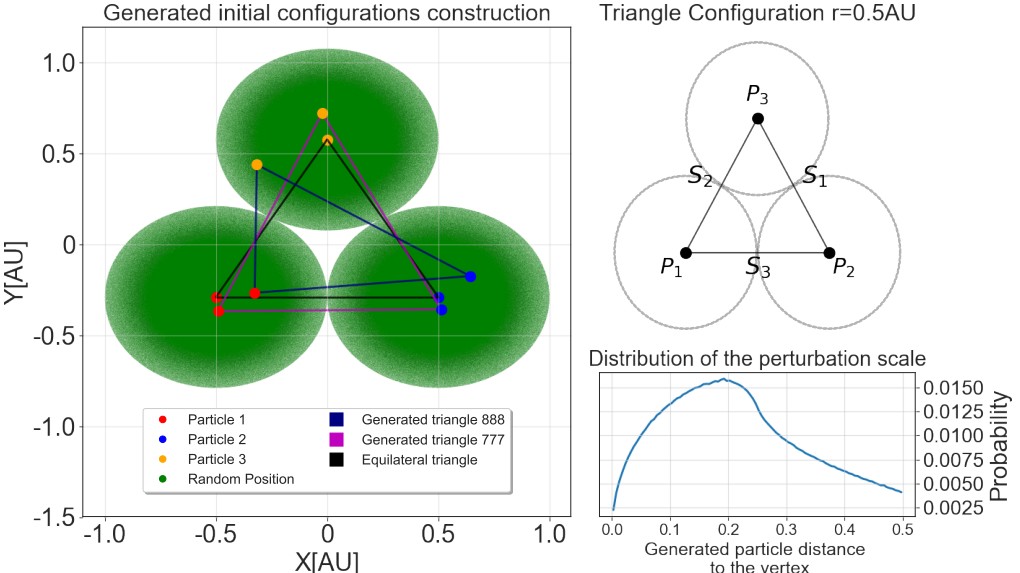

Figure 2: Left panel: three perturbed triangles configurations. The green circles indicate the boundaries where each new particle can be generated. Top-right panel: initial state of the system corresponding to an equilateral triangle 1 au side. Bottom-right panel: the distribution of perturbation radius, defined as the distance from the perturbed position to the vertex of the triangle.

Table 1: For **Set A** and **Set B**, the initial particle positions are sampled following the procedure described in Section (3.2.2). For **Set C**, the initial configurations are shown in Section (3.4). The variable $r_s$ corresponds to the Schwarzschild radius for a $1M_\odot$ particle.

|  | Number of Simulations | Particle radius | Vertex radius [au] |
|---|---|---|---|
| Set A | $10^7$ | Point particle | 0.5 |
| Set B | $5 \times 10^6$ | Point particle | $10^{-7}$ |
| Set C | $5 \times 10^5$ | $500_{r_s}$ | 0.5 |

As shown in Figure 1, for every new triangle that we generate, we re-compute the lengths of each side, where $S_1$ $S_2$ and $S_3$ represent the initial lengths of the sides opposing particles 1, 2, and 3, respectively. These choices are made to set a good balance between drop-in time and the total number of simulations we can perform (i.e., shorter drop-in times result in simulations that finish quicker, hence we can perform more simulations for a constant total real-time run-time), and to connect each one of the possible initial configurations to the properties of the ejected particle and the left-over binary (see Figure 3 and Figure 7).

## 3.3 Ejection criteria

To determine when a particle is ejected from the system, we check the state of the system at every time step (each simulation has a maximum integration time of $10^5$ yrs). The most bound pair is defined as the binary, and at every time-step, we check if the ejected single remains bound to the binary centre of mass. The integration stops if the escaping single star has positive total energy and the final hyperbolic binary-single orbital separation is 100 times that of the most compact binary semi-major axis.

Roughly 0.12% of our simulations in **Set A** did not finish, i.e the integration time needed for some systems to fulfill our ejection criteria is longer than our total integration time. This is typically due to long-lived Lévy flights, where the return time of the escaper exceeds our chosen maximum integration time. This represents a negligible fraction of our simulations, and our results remain unchanged without these simulations.

## 3.4 Simulations with post-Newtonian corrections

We use **Set A** to define our initial particle positions, but set the radius of each particle to be equal to 500 times the Schwarzschild radius. Particles are assumed to merge using the sticky-star approximation: if the particle radii overlap in both time and space, we assume a merger occurs. We choose 500 Schwarzschild radii for the following reasons. First, when two particles are closer than this distance, they are practically decoupled from the rest of the system, and the binary will merge very quickly. Second, the post-Newtonian approximation used to correct the Newtonian acceleration begins to break down at around this distance [45]. Third, by avoiding integrating the final part of the inspiral we save a great amount of computational run-time. This is because, as the period of the merging binary decreases, the post-Newtonian corrections would start to dominate and the integration time would increase. This scenario is avoided by choosing $500r_s$ for the particle radius and by adopting the sticky-star approximation to determine when the particles

merge (i.e., when their radii overlap in both time and space)

## 4   Theoretical Expectations

In this section, we describe the theoretical expressions (see Chapter 7 of [46]) for the expected distributions of the final binary orbital energy or, equivalently, the final binary semi-major axis (i.e.,the inverse binding energy distribution), the final binary eccentricity, the single star ejection velocity and total system lifetime, for comparison to our numerical scattering results.

### 4.1   Dividing the phase space into ergodic and regular subsets

In order to properly compare our simulated data to theoretical expectations, the simulations must first be divided into chaotic and regular subsets. This is done using the criterion defined in Section 2.3. In Section 5, we will perform our comparisons with and without applying these cuts, to better understand how the chaotic and regular subsets are each contributing to the total distribution (i.e., without applying any cuts for ergodicity).

### 4.2   Orbital energy distribution

We use the inverse orbital energy distribution we obtain from our simulations to compare to the theoretical binary orbital energy distributions provided in [46] and shown in Figure. 6. As we are working with particles released from rest, the initial kinetic energy is equal to zero. We can thus write the initial energy in terms of the potential energy alone, and the binary binding energy in terms of the binary semi-major axis:

$$E_0 = -G\left(\frac{m_1 m_2}{|\mathbf{r}_1 - \mathbf{r}_2|} + \frac{m_1 m_3}{|\mathbf{r}_1 - \mathbf{r}_3|} + \frac{m_2 m_3}{|\mathbf{r}_2 - \mathbf{r}_3|}\right) \tag{3}$$

$$E_B = -G\left(\frac{m_a m_b}{2a}\right), \tag{4}$$

The variables $m_a$ and $m_b$ correspond to the particle masses in the final left-over binary post-interaction.

The distribution used in [46] for the inverse orbital energy is given by Equation 5, where $z = |E_0|/|E_B|$ corresponds to the inverse binary orbital energy. For the planar case, this is

$$f(z) = 3.5z^{2.5}, \tag{5}$$

### 4.3   Eccentricity distribution

We compare our simulated results to several different theoretical eccentricity distributions in Figure. 6. These are described below.

The thermal distribution for the binary eccentricity, which assumes a detailed steady-state balance between binary creation and destruction [47], is given by Equation 6 :

$$f(e)de = 2ede, \tag{6}$$

In the planar case, the eccentricity should be distributed as Equation 7

$$f(e)de = e(1 - e^2)^{-1/2}de, \tag{7}$$

The power-law index on $(1 - e^2)$ is taken to be $-1/2$, which is appropriate for the low angular momentum limit [8, 48].

## 4.4 Escape velocity distribution

We adopt Equation 7.19 in [46] for the escape velocity distribution shown in Figure. 6. This is appropriate to the planar case and is given by

$$f(v_s)dv_s = \frac{(3.5|E_0|^{7/2}(m_sM/m_B))v_sdv_s}{(|E_0| + \frac{1}{2}(m_sM/m_B)v_s^2)^{9/2}}, \tag{8}$$

# 5 Results

In this section, we present the results of our numerical scattering experiments. We begin by presenting the phase space plots for our zero angular momentum equilateral triangle set-up, before moving on to the final calculated distributions for the post-interaction binary orbital energy, orbital eccentricity, escaper velocity and total system lifetime. We then compare these to the expected theoretical predictions, using our phase space plots to identify the origins of the disagreements between our theoretical calculations and the simulated data.

## 5.1 Phase space divided by ejected particle identity

We represent the initial phase space in 2 dimensions, so that each point in this space corresponds to a unique initial configuration (i.e. a single simulation). For our purpose, the parameters chosen for the $X$ and $Y$ axes are the ratios between the lengths of the sides of each of the generated triangles, as shown in Figure. 2. Here, $S_1$, $S_2$ and $S_3$ correspond to the lengths of the sides opposing, respectively, particles $P_1$, $P_2$, and $P_3$. We then plot the ratios $S_1/S_2$ and $S_1/S_3$ on each axis to construct our phase space plots for the outcomes of the perturbed equilateral triangle experiment. In this coordinate system, the singularity is located at the coordinates $(1, 1)$, and corresponds to the initial conditions for a perfect equilateral triangle.

We colour-code each data point, using the colour associated with the ejected particle (see Figure. 3). In this case, uniform swaths of colour correspond to prompt regular outcomes and multi-coloured patches correspond to chaotic regions of phase space.

### 5.1.1 Full phase space

The central panel in Figure 3 shows the phase space limited to $0 < S_1/S_2 < 3$ and $0 < S_1/S_3 < 3$ for the interactions of **Set A**.

Near the singularity, we see regular swaths of uniform colour, separated by chaotic regions of phase space. In the regular regions, small perturbations to the initial state do not lead to a different particle being ejected. In the multi-coloured regions of our phase space corresponding to chaotic or ergodic regions, the identity of the ejected particle is very sensitive to the initial

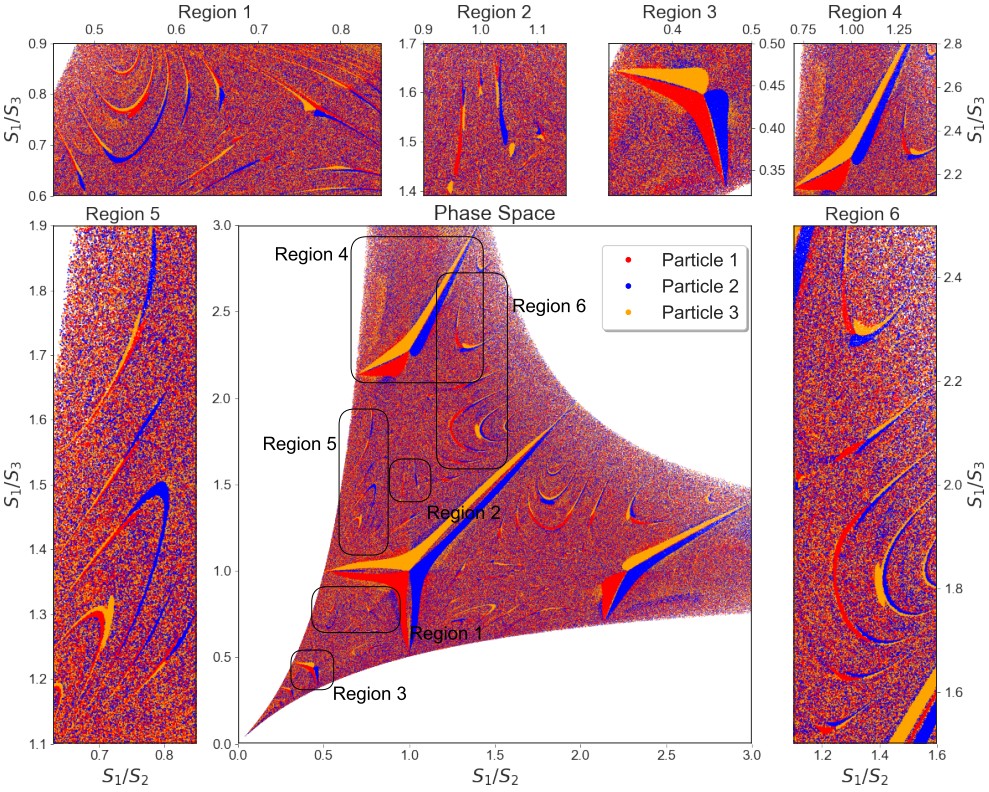

Figure 3: Phase space plot, color-coded by ejected particle. X- and y-axes indicate the ratio between the sides $S_1/S_2$ and $S_1/S_3$ of the perturbed equilateral triangles. The singularity is at $S_1/S_2 = S_1/S_3 = 1$, corresponding to a perfect equilateral triangle. Each subsequent inset corresponds to a zoom-in on the regions in the phase space.

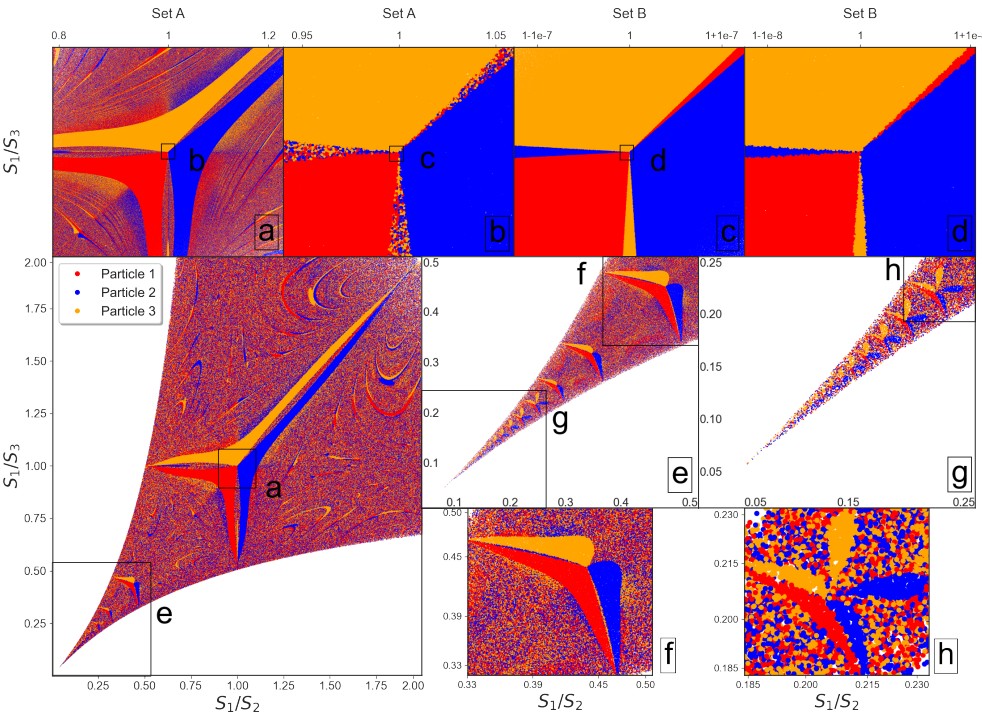

Figure 4: Same as Figure 3, but zoomed on specific regions of the phase space. Top panels, left to right: each subsequent inset on the top is a zoom-in on the singularity (see panels a,b,c, and d). Bottom panels e and g zoom on the regime of binary-single scattering, where one side is very small compared to the others. Bottom right panels f and h: the zoomed-in version of the two different shapes we see emerging in panels e and g.

conditions, causing different particles to be the ones ejected in spite of very minor changes to the initial conditions.

Moving away from the central singularity, we see emerging more and more chaotic regions of phase space. Moving far from the singularity to initial conditions corresponding to extremely deformed triangles, the phase space has three arms. In each arm, the initial configuration is approaching the binary-single scattering regime, since the length of one of the sides of the triangle is very small. The arm that goes to the top represents the deformation of the triangle where particle $P_3$ (coloured by orange) is initial far from particles $P_1$ and $P_2$. Similarly, the arms that goes to the right and the left bottom represent the deformation of the triangle where the particles $P_2$ (coloured by blue) and $P_1$ (coloured by red) are far from the others.

### 5.1.2 Zoom-in on the singularity

In **Set B**, the perturbations to the equilateral triangle are very small. Therefore, we can better appreciate the behaviour of the system very close to the three-body collision.

Each top panel in Figure. 4 is a subsequent zoom-in on the central singularity, with the zoom-ins becoming more extreme going from left to right. We see regular coloured regions with clear

boundaries separated from the chaotic ones. As we zoom-in more and more on the central singularity, we find that the chaotic regions disappear to eventually become fully regular in the vicinity of the singularity.

Panel **a** is a zoom-in on the singularity for **Set A**. We see regular uniformly coloured regions with clear boundaries separated from the chaotic ones. In panel **b** (a subsequent zoom-in on the singularity corresponding to panel **a**) we see a plume of uniform colour emerging in between the red and yellow swaths of regular regions of uniform colour. This blue plume becomes prominent as we zoom-in more, until eventually we are left with only uniform colouring in the vicinity of the singularity as it is shown in panel **d**. Here we find that the chaotic regions disappear and the phase space becomes fully regular in the vicinity of the singularity.

### 5.1.3 Zoom-in on the binary-single regime

The bottom left panel of Figure 4 shows the same phase space as Figure 3 of our **Set A** , but for smaller values of $S_1/S_2$ and $S_1/S_3$. The right middle pair of panels shows the binary-single scattering regime. The bottom right figures show two zoom-in on the regular regions of the binary-single scattering regime.

The three branches of the "triangle" correspond to the binary-single scattering regime, where two of the bodies are very far from the third. In panel **e**, we see regular regions repeating at approximately regular intervals, which are separated by chaotic patches of phase space. The first three regular regions have roughly the same shape (see panel **f**). However, the second zoom-in panel reveals two bulges arising from the centre of each regular region (see panel **g**). The bulges increase in size as we move farther into the extreme binary-single scattering regime (see panel **h**). Importantly, the regular structures do not disappear and appear to continue indefinitely down to increasingly small scales.

### 5.2 Including post-Newtonian corrections and general relativistic effects

Very near to the singularity, general relativity and quantum mechanical effects should begin to become important as the distance between the particles approach the Planck scale. We re-perform our suite of scattering experiments, adopting the same sets of initial conditions, but now including post-Newtonian corrections to account for the effects of general relativity. The results of this exercise are shown in Figure 5, which shows how Figure 4 changes when general relativity is accounted for.

We find that turning on general relativity (i.e turning on post-Newtonian corrections) causes us to lose resolution near the singularity. This is made evident by the light-grey data points, which correspond to prompt mergers of two of the particles. We also find that we lose resolution in the binary-single scattering regime. Due to the loss of resolution near the singular regions, we can no longer verify that the previously identified patterns (e.g., repeating swaths of chaos and regular regions) will continue down to the Planck Scale, assuming gravity remains classical in this limit. This loss of resolution occurs because the two black holes merge when they approach the Schwarzschild radius, which is much larger than the Planck scale.

### 5.3 Comparison with theoretical expectations

In this section, we present our final simulated distributions of binary orbital energies, orbital eccentricities, escaper velocities and total system lifetime, and compare the results to the theoretical

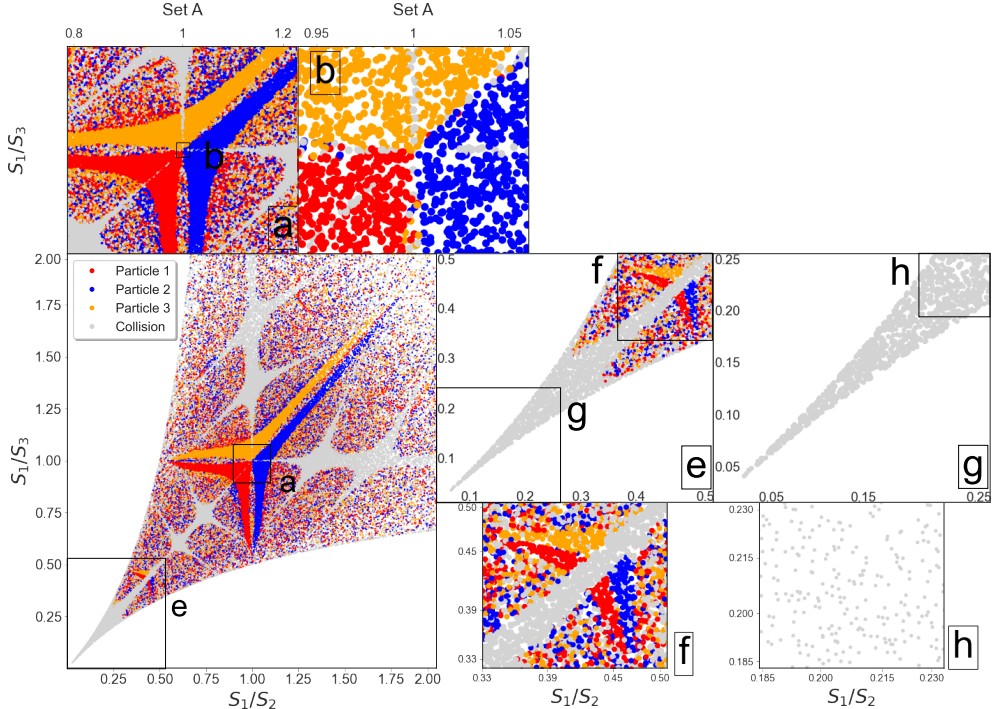

Figure 5: Same as Figure 4, but employing post-Newtonian corrections. The light grey dots correspond to simulations ending in a two-body collision, or merger.

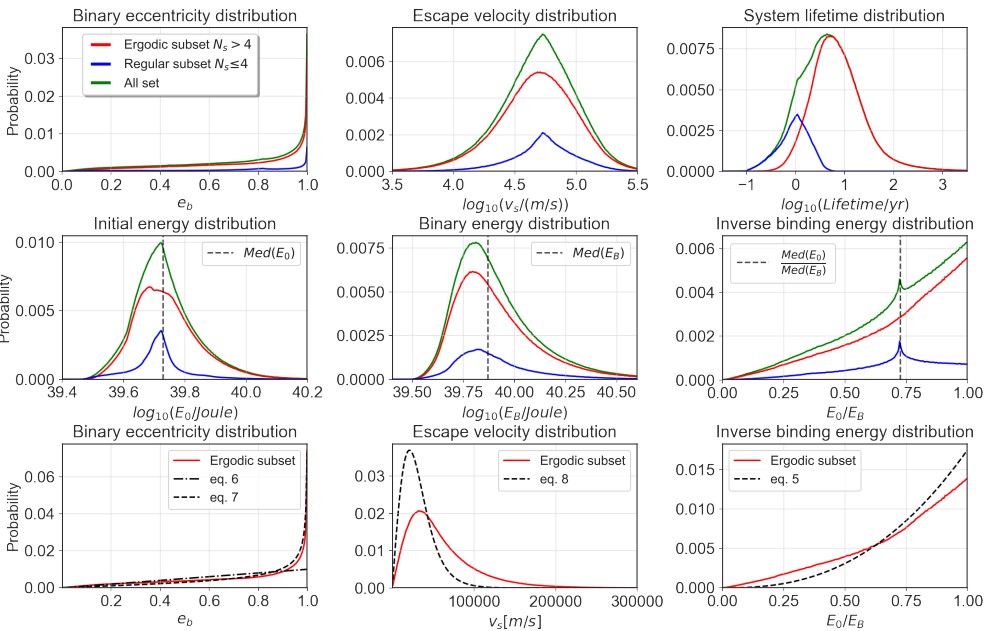

Figure 6: Distributions of outcome parameters. The green lines show the distributions for our entire suit of experiments, blue lines correspond to the regular subset where $N_S \leq 4$, the red line correspond to the ergodic subset where $N_S > 4$. Bottom panels show the ergodic subset along with the theoretical curves shown in 4.

expectations provided in Section 4. Because the statistical theory is based on purely Newtonian physics, we compare it only with our **Set A**, which does not include post-Newtonian corrections.

### 5.3.1 Distributions of outcome properties

In the top panels of Figure 6, the histograms show the final distributions of binary eccentricities, ejected particle velocities, and system lifetimes. The distributions are separated into regular (all simulations with $N_S \leq 4$) and ergodic (all simulations with $N_S > 4$) subsets, along with the entire suite of simulations in **Set A**. The criterion for separating the two subset is $N_S \geq 4$, as discussed in Section 2.3.

In the middle panels of Figure 6, the histograms show the distributions of initial system energies, binary orbital energies, and the inverse binding energies. Relative to the theoretical expectation, we find too many compact binaries and obtain not enough wide binaries.

In the bottom panels of Figure 6, the histograms show the distributions of binary orbital eccentricities, ejected particle velocities and inverse binding energies, respectively, and compare them with our theoretical expectation (see the equations in Section 4). Our results produce an excess of eccentric binaries relative to a thermal distribution (Equation 6), but agree overall quite well with the zero angular momentum distribution (Equation 7). Relative to the theoretical expectation for the ejection velocities (Equation 8), we obtain an excess of high-velocity escapers and a lack of low-velocity escapers.

### 5.3.2   Phase space distribution of outcome properties

Figure 7 shows the initial phase space plots colour-coded by the properties of the final particles. The use of the initial phase space plots is particularly useful to identify the regular regions, which are likely the cause of the disagreement between the simulations and the statistical theory.

In the zero angular momentum limit, our naive expectation is to preferentially produce compact binaries with high eccentricities (i.e., binaries with low orbital angular momentum). But this is not always observed in our experiments. This occurs because the escaper tends to carry away some significant angular momentum, requiring the binary to retain the remaining reservoir of angular momentum (but going in opposite directions in order to cancel out to zero).

This is clearly observed in Figure 7 upon comparing the ejection velocities to the orbital properties of the binary.

Focusing on the exterior boundaries of the regular regions surrounding the central singularity, we observe higher ejection velocities corresponding to higher eccentricities and wider binaries. Conversely, focusing on the interior boundaries of the same regular regions, we see higher ejection velocities correlating with low eccentricities and more compact binaries. Focusing on the singularity, we observe compact binaries correlated with high eccentricities and high ejection velocities. In the limit of the escaper velocity becoming very low, we tend to observe either high eccentricities and wide binaries or low eccentricities and compact binaries, since the escaper is no longer able to carry away much angular momentum, or $L_S \sim 0$. Conversely, as the escaper velocity reaches its maximum near the singularity, we observe more compact binaries and higher eccentricities, since the binary is left with little to no remaining angular momentum. It is for these reasons that we observe a supra-thermal eccentricity distribution and a binary orbital energy distribution that prefers compact binaries.

The shortest-lived systems are closest to the singularity, with the external borders of the main regular regions reaching the longest lifetimes, and correlating with high ejection velocities, high eccentricities, and compact binaries. Very near to the singularity, the very short interaction lifetimes likely indicate regular interactions, which is consistent with the progressive zoom-in on the singularity shown in Figure 4. The low lifetime regions are surrounded by a narrow strip of very long Lévy flight interaction, consistent with the results of [38].

### 5.3.3   Weak chaos and the disagreements between theory and simulations

In this section, we summarize our findings regarding the disagreements between our theoretical expectations and simulated results, as presented in the previous sections. This is done in an effort to identify their origins using the post-interaction distributions of binary orbital energies, orbital eccentricities and escaper velocities. In this section, we divide our total suite of simulations into ergodic and regular subsets. To do this, we select our data following the the scramble number criterion discussed previously in Section 2.3 ($N_S > 4$), and isolate most of the regular regions from the chaotic ones. Approximately 17.9% of the simulations belong to the regular regions and 82.1% belong to the ergodic regions, revealing that most of the parameter space is chaotic.

One of the main inconsistencies of the data with our theoretical expectations is the bump at z = 0.71 in the inverse binding energy distribution (see the right-hand middle panel of Figure. 6). In order to identify the origin of this bump, we plot separately the distributions for the ergodic and regular subsets for both $E_0$ and $E_B$. We find a misalignment between the peaks of the regular $E_0$ and $E_B$ distributions relative to the medians (see middle insets of Figure. 6). The origin of this misalignment can be traced back to the slightly asymmetric distribution of initial binary orbital

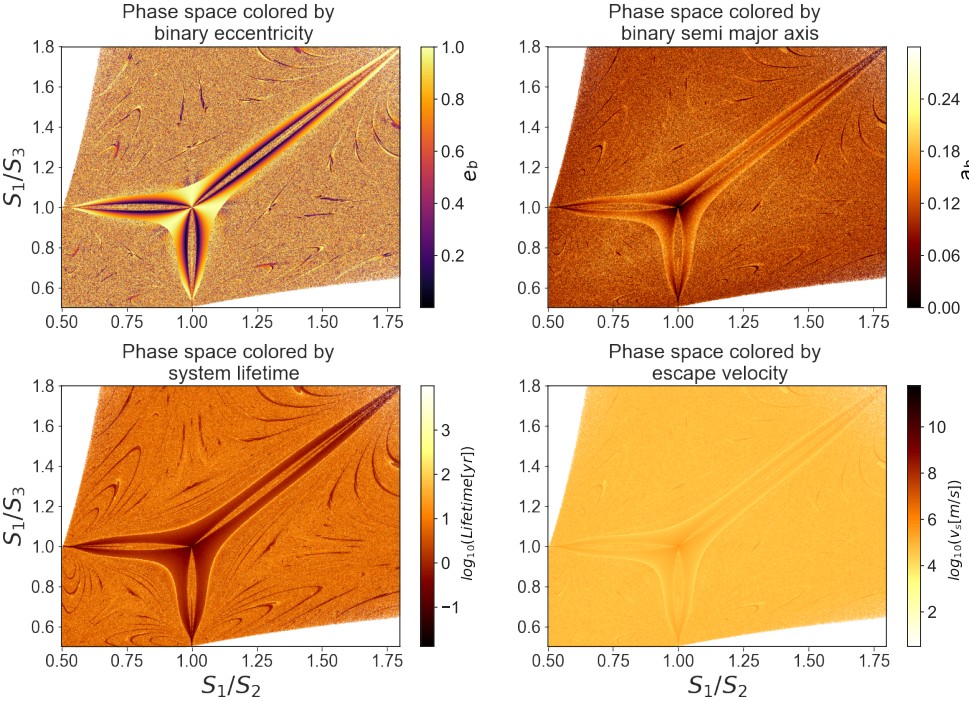

Figure 7: Color-coded phase space plot showing the final binary semi-major axes (top left inset), orbital eccentricities (top right inset), escaper ejection velocities (lower right panel) and total system lifetimes (lower left panel).

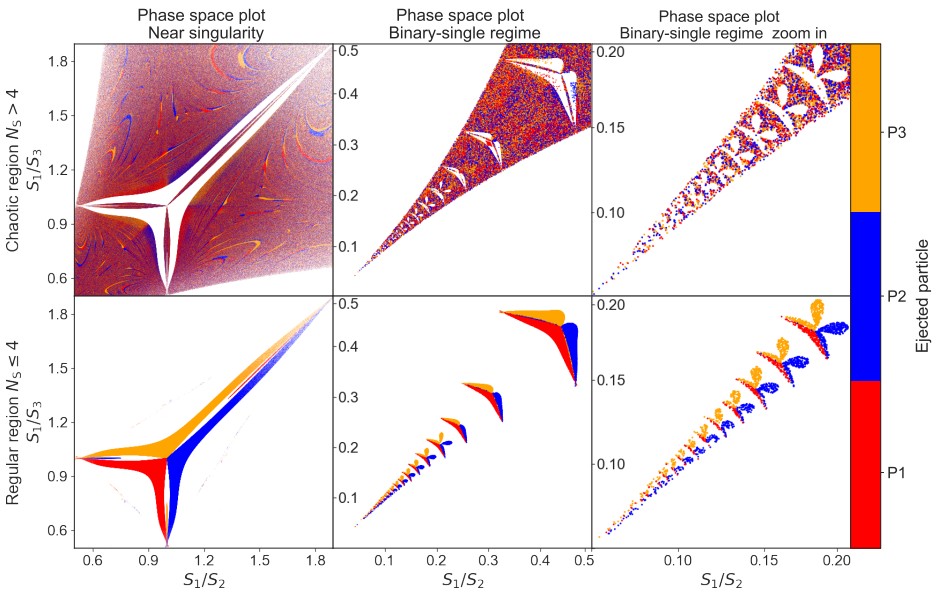

Figure 8: Top panels: phase space for the ergodic subset near the singularity (left) and in the binary-single scattering regime (middle and right). The bottom panel shows the same regions of the top panels, but for the regular subset instead.

energies from which we sampled our initial conditions, which slightly over-populate the hard end of the distribution relative to the soft end.

As a consequence of this misalignment, we expect a bump in the regular subset of the inverse binding energy distribution, as seen in the right-most middle panel of Figure. 6. In fact, the ratio between the medians of the $E_0$ and $E_B$ distributions matches the peak in the inverse binding energy distribution for the entire suit of simulations, therefore confirming that the origin of the misalignment is in the regular, and not in the ergodic, subset.

### 5.3.4 Ergodic and regular phase space

We can better appreciate the clear distinction between regular regions and the chaotic regions by separating them in the initial phase space plots of Figure. 3.

Figure 8 shows the region of our phase space close to the singularity for which $S_1/S_2 = S_1/S_3$ (left column) and the binary-single regime ($S_1 \gg S_2, S_3$. middle and right panels). The last two panels show a zoom-in of the middle panels. The top row shows the ergodic subset, and the bottom row shows the regular subset. As expected, our scramble criterion neatly separates the regular and chaotic regions, including the triangle-shaped regions around the singularity and the self-repeating structures in the binary-single regime.

# 6 Discussion & Summary

In this paper, we consider an idealized initial configuration of the general three-body problem, in order to study chaos in the vicinity of a singularity in the general N-body problem. The experiment we perform is to construct an equilateral triangle, with the initial positions of the three identical particles located at each vertex. For a perfect equilateral triangle, all three particles will eventually collide at the centre of the triangle in a three-body collision, if released from rest at the same time and all particles have equal masses. This would give rise to a singularity in the Newtonian acceleration because the particles would occupy the same position in space and time at the system centre of mass. By perturbing the triangle and re-performing the experiment, the particles arrive at the origin slightly offset in space and time. The larger the perturbations, the farther is the parameter space from this singularity.

We perform in total $10^7$ simulations using the TSUNAMI code [37]. The large suite of simulations proves necessary in order to generate phase space plots with sufficient resolution to be interpretable (i.e., the chaotic and regular zones of the parameter space are clearly identified). With our final suite of numerical simulations, we generate the final distributions of binary orbital energy, orbital eccentricity, escaper velocity, and total interaction lifetime. We then compare these distributions to the standard theoretical expectations provided in the literature. We find significant differences in the final orbital properties of our binaries, relative to the theoretical expectations, including more eccentric orbits and a higher fraction of more compact binaries. Using our phase space plots, we explain the origins of these differences, which primarily come down to the total angular momentum carried away by the escaping single star. For example, in the limit of $v_\infty \rightarrow 0$, the left-over binary tends to be more circular, since it contains the entire angular momentum reservoir of the interaction.

Finally, using our results for this idealized experiment in the zero-angular momentum limit, we make predictions for the expected properties of binaries formed via three-body interactions of initially isolated single stars in very dense environments. We expect these predictions to be most applicable to star clusters with isotropic stellar velocity distributions and little to no rotation.

## 6.1 Theoretical implications for the three body problem and quantum gravity

In this section, we discuss the implications of our results for the general three-body problem and quantum gravity. This is because we are perturbing our experiments in the vicinity of a singularity in the three-body problem, and our simulations extend to very small spatial scales.

### 6.1.1 Chaos Below the Planck scale?

Our suite of simulations are designed to include very small perturbations of the initial conditions in the vicinity of both singularities in our idealized three body experiment. These are the singularity for a perfect equilateral triangle (i.e., $S_1/S_2 \sim S_1/S_3 \sim 1$, and the singularity in the binary-single scattering regime (i.e., $S_1/S_2 \sim S_1/S_3 \ll 1$). Both singularities correspond to limits where two or more particles are overlapping in space and time, creating a singularity in the gravitational acceleration. In principle, our simulations allow us to analyze our results all the way down to very small scales, possibly extrapolating down to scales of order the Planck length and beyond. As described below, we see interesting behaviour in the vicinities of the singularities in this idealized three-body experiment. But can we reliably extrapolate our results into these regimes? As we will explain, the complication that arises is that the scales extend to such small perturbations, that

both quantum mechanical and general relativistic effects begin to become important, reducing our resolution in the vicinities of the singularities.

### 6.1.2 Chaos in the vicinity of the singularity for a perfect equilateral triangle

In the top panels of Figure 4, we see a zoom-in of the singular region near $S_1/S_2 \sim S_1/S_2 \sim 1$. We see three primary regular regions, corresponding to the ejections of each of our three particles. These islands of regularity are separated by chaotic regions of phase space. However, as we zoom-in on the singularity, we see these chaotic patches disappear, and the phase space becomes fully regular at very small scales. This is shown in the insets to Figure 4 which show, from left to right, progressive zoom-ins. With each successive zoom-in, the chaotic regions of phase space are diminished, until they disappear completely at sufficiently small scales (see the top right-most inset).

Is this result telling us that gravity remains classical and fully regular at very small scales? Unfortunately, this question becomes difficult to address once general relativistic corrections have been implemented. As shown in Figure 5, turning on post-Newtonian corrections washes out our resolution in the vicinity of the singularity, due to rapid mergers of two particles that begin very close together. Consequently, we are unable to apply our results at these scales with confidence, and this prohibits us from drawing any firm conclusions about quantum gravity from our experiments at this time.

### 6.1.3 Chaos in the binary-single scattering regime

In the bottom panels of Figure 4, we see a zoom-in of the binary-single scattering regime where $S_1/S_2 \sim S_1/S_3 \ll 1$. We see islands of regularity repeating at nearly regular intervals, separated by patches of chaos. This behaviour seems to continue indefinitely to extremely small scales, with the shapes of the islands of regularity changing progressively as we approach the limit $S_1/S_2 \sim S_1/S_3 \rightarrow 0$.

If we extrapolate this result down to smaller scales, is it telling us that gravity remains *both* regular *and* chaotic below the Planck scale? As before, this question becomes difficult to address once post-Newtonian corrections are turned on. In this limit, we lose resolution due to two particles merging almost instantaneously at the beginning of our simulations, since they are so close in space that general relativity predicts an instantaneous merger. Thus, as before, we are unable to apply our results at these scales with confidence, prohibiting us from drawing firm conclusions about quantum gravity at this time. The question now becomes: *How* can we use our suite of simulations be used to extrapolate our results down to the Planck scale and beyond?

In future work, we hope to improve upon this limitation of our work. As a first step, we will adapt the experiment such that the triangle is initially rotating, and balance the centripetal and gravitational forces acting on each particle. This creates a singularity in time, since all three identical particles should take an infinite amount of time to reach the origin for a perfectly rotating equilateral triangle. This increases the angular momentum reservoir, and will allow us to understand and even quantify how the total system angular momentum affects the fraction of the phase space corresponding to chaos and regularity.

Another pathway to improve our experiments is to introduce some probabilistic element into the evolution of the particles, in order to mimic the effect of Planck-scale perturbations and the uncertainty principle. This would however require us to change the integration algorithm from a leapfrog to a quasi-symplectic integrator, suitable for the evolution of stochastic differential

equations. But, in principle, it should be possible to perform integrations where we manage the error accumulation with extreme accuracy, and using our uncertainties from the simulations, can explore the limit at which the errors due to the computational simulations add up to violate the uncertainty principle, and the exact point at which quantum mechanical effects should prohibit us from extrapolating to small scales.

## 6.2 Astrophysical implications

In this section, we discuss the properties of binaries formed via three-body interactions in dense environments, and compare our results to theoretical expectations for the ergodic subset of our simulations.

### 6.2.1 Three body binary formation in isotropic clusters

In isotropic star clusters, binaries formed from three-body interactions of initially all single stars should have a low total angular momentum. This is because there is little to no angular momentum in the cluster, and the stellar orbits are such that three single stars that meet near the cluster centre of mass should be on nearly radial orbits, causing the three stars to approach their common centre of mass with small impact parameters. In the opposite limit of high angular momentum and three-body binary formation in clusters with significant rotation, we would expect the opposite. That is, the three stars should all approach their common centre of mass with large impact parameters, causing the total angular momentum to be large, since the orbits tend to be less radial and more tangential due to rotation. In this paper, we focus on the zero angular momentum limit, and defer the rotating case to a future paper.

### 6.2.2 Binary properties

For the ergodic subset, stars are ejected at higher velocities relative to the theoretical expectation. Hence, the single star can take away a significant fraction of the total angular momentum reservoir. In this limit, we produce more wide binaries relative to the theoretical expectation, along with very high eccentricities. In the limit that the single star takes away no angular momentum (i.e., its final velocity at infinity is zero), we find more compact binaries and smaller eccentricities relative to the theoretical expectations.

Overall, our results suggest that binaries formed from three-body interactions in isotropic star clusters should end up with orbital parameters that correspond to rapid coalescence for binaries composed of two black holes due to the emission of gravitational waves (GWs). We tend to find either compact circular binaries or wide very eccentric binaries. In both cases, the merger times will be short, since the timescale for a merger to due gravitational wave emission is proportional to the binary orbital separation $\propto a^4$ (i.e., more compact binaries merge faster) and also proportional to $(1 - e^2)^{7/2}$ for the orbital eccentricity [49].

# Acknowledgements

N.W.C.L. gratefully acknowledges support from the Chilean government via Fondecyt Iniciacion Grant 11180005, and acknowledges financial support from Millenium Nucleus $NCN\,19-058$ (TI-TANs). A.A.T. acknowledges support from JSPS KAKENHI Grant Numbers 17H06360, 19K03907 and 21K13914.

## Data Availability

The TSUNAMI code, the initial conditions and the simulation data underlying this article will be shared on reasonable request to the corresponding author.

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
