# Peer review of "Chaos in the vicinity of a singularity in the Three-Body Problem: The equilateral triangle experiment in the zero angular momentum limit"

_SciPost Physics Core_

## Round 2 · Referee Report · Anonymous (Referee 1) · 2022-5-8

Strengths

Authors present the problem they are studying in a clear and concise language. They also present their methodology and results in a manner which is reproducible.

Weaknesses

The paper focuses on a simple three body system, but does not explore in detail any astrophysical application. While the numerical results are new, the paper lacks new theoretical results to explain discrepancy between numerical results of this work and theoretical results from literature. In addition, certain sections of the paper may benefit from additional discussion and explanation.

Report

The authors study the evolution of a system of three particles initially positioned on the corners of an equilateral triangle with zero initial velocity. If positioned on a perfect equilateral triangle, the system would encounter a singularity. In this study the authors perform N-body simulations on a perturbed system and study outcomes of such simulations in a statistical manner. They then compare the results of the simulations with other results in literature.

Requested changes

  1. The paper might benefit from an additional figure with trajectories of the three point objects from one of the runs in the ensemble of simulations shown in Figure 3. This might illustrate the prevalence of chaos in the system. This may also be used to illustrate the importance of numerical chaos as discussed in Section 1.

  2. Since the code used in this paper(TSUNAMI) has not been published yet, it might be useful to include a figure which compares the results of this code with other popular N-body integrators used in literature. The authors may highlight how and where TSUNAMI works better than other codes.

  3. Page 7 : It is not clear to me as to why the radial perturbation is samples uniformly in $r^2$ and not r.

  4. Figure 3: Away from the singularity, there seems to be regions of regularity embedded in the chaotic regions. It would be useful if the authors could explain what cause of these regular regions.

  5. Page 14 and Figure 5: It is not clear as to why adding GR effects would reduce the resolution in Figure 5. It seems that adding GR would convert some of the red,blue and orange dot to grey ones.

  6. Figure 5: It would be useful if the authors used a color other than grey to plot mergers. It is difficult to identify them against the white background, especially in the zoom-ins (e.g. zoom in b).

  7. Page 17: Despite the fact that the theoretical predictions are for Newtonian gravity, it would be useful to compare them with statistical results of Set C. This way authors could highlight the differences GR would cause.

  8. Figure 7, Bottom right panel: It is difficult to discern the different regions of phase space. It would be helpful to use a different color scheme or normalization.

  9. Figure 6,7: It might be useful to report semi-major axis and in dimensionless qualities rather than in S.I. units. That way the results could be easily scaled for other systems.

  10. Page 17: It would be useful to quantify various adjectives used while describing the results. For instance it would to useful to quantify what the authors mean by wide binaries.

11.Page 17: Do your results change by changing the criteria for ergodicity? For instance would a choice of $N_s>2$ significantly change the comparison with theoretical predictions?

  1. Section 6.1.1 : It is not clear why authors are discussing quantum mechanical effects in the discussion section when the analysis was based on purely Newtonian and Relativistic calculations.

  2. Sections 6.2.1-6.2.2: It would be useful if the authors could elaborate on these sections with some order of magnitude calculations.

---

## Editorial Decision

resubmitted